# SHINRA2020-ML: Categorizing 30-language Wikipedia into fine-grained NE based on "Resource by Collaborative Contribution" scheme

**Satoshi Sekine**                                   SATOSHI.SEKINE@RIKEN.JP
**Kouta Nakayama**                               KOUTA.NAKAYAMA@RIKEN.JP
**Maya Ando**                                       MAYA@KZD.BIGLOBE.NE.JP
**Yu Usami**                                                   YU@USAMI.LLC
**Masako Nomoto**                             MASAKO.NOMOTO@RIKEN.JP
**Koji Matsuda**                                     KOJI.MATSUDA@RIKEN.JP
**Asuka Sumida**                                   ASUKA.SUMIDA@RIKEN.JP
*1-4-1 Nihonbashi, Chuo-ku, 15th floor*
*Tokyo, 103-0027, Japan*

## Abstract

This paper describes a Knowledge Base construction project, SHINRA and particularly the SHINRA2020-ML task. The SHINRA2020-ML task is to categorize 30-language Wikipedia pages into fine-grained named entity categories, called "Extended Named Entity (ENE)". It is one of the three tasks conducted on SHINRA since 2017. SHINRA is a collaborative contribution scheme that utilize Automatic Knowledge Base Construction (AKBC) systems. This project aims to create a huge and well-structured knowledge base essential for many NLP applications, such as QA, dialogue systems and explainable NLP systems.

In our "Resource by Collaborative Contribution (RbCC)" scheme, we conducted a shared task of structuring Wikipedia to attract participants but simultaneously submitted results are used to construct a knowledge base. One trick is that the participants are not notified of the test data, so they have to run their systems on all entities in Wikipedia, although the evaluation results are reported for only a small portion of the test data among the entire data. Using this method, the organizers receive multiple outputs of the entire data from the participants. The submitted outputs are publicly accessible and are applied to building better structured knowledge using ensemble learning, for example. In other words, this project uses AKBC systems to construct a huge and well-structured Knowledge Base collaboratively.

The "SHINRA2020-ML" task is also based on the RbCC scheme. The task categorizes 30-language Wikipedia pages into ENE. We previously categorized the entire Japanese Wikipedia entities (920 thousand entities) into ENE by Machine Learning and then checked by hands. For SHINRA2020-ML task participants, we provided the training data using categories of the Japanese Wikipedia and language links from the page. For example, out of 920K Japanese Wikipedia pages, 275K have language links to German pages. These data are used to create the training data for German and the task is to categorize the remaining 1,946K pages. We conducted a shared task for 30 languages with the largest active users, and 10 groups participated. We showed that the results by simple ensemble learning, i.e., majority voting, exceed top results in 17 languages, thereby proving the usefulness of the "RbCC" scheme.

We are conducting two tasks in 2021, SHINRA2021-LinkJP and SHINRA2021-ML tasks. We will introduce these tasks in a later section of the paper.

## 1. Introduction

Wikipedia consists of large volumes of entities (articles), which are great resources for knowledge used in many NLP tasks. To maximize the use of such knowledge, resources created from Wikipedia are structured for inference, reasoning, or any other purposes in many NLP applications. Several machine readable knowledge bases exist, including Cyc [Lenat, 1995], DBpedia [Lehmann et al., 2015], Yet another greater ontology (YAGO) [Mahdisoltani et al., 2015], Freebase [Bollacker et al., 2008], Wikidata [Vrandečić and Krötzsch, 2014] and so on, each of them has problems to be solved. Recent KBs are created in a bottom-up manner by crowdsourcing, causing a significant amount of undesirable noises in the knowledge base. We believe that the structure of the knowledge should be defined in a top-down manner rather than a bottom-up manner to create cleaner and more valuable knowledge bases. Instead of existing cumbersome Wikipedia categories, we rely on a well-defined and fine-grained category. Extended Named Entity (ENE) [ENE-HP] is one of the well-defined name ontologies with 219 hierarchical categories and a set of attributes is defined for each category. We employed ENE for as the structure of the knowledge base in this study.

The automatic knowledge base construction (AKBC) and document categorization shared task have been popular for decades. These popular shared-tasks are common in the field of Information Retrieval, Information Extraction, Knowledge Base population, and attribute extraction, such as TREC, MUC, ACE, KBP [U.S. National Institute of Standards and Technology (NIST) , 2018] and CoNLL. However, most of these tasks are designed only to compare the system performance and find the best ranked systems on limited test data. The outputs of the participating systems are not shared and the systems may be abandoned once the task is over.

Our contribution is geared to improve the situation by the following changes:

1. Designing the shared-task to construct a knowledge base rather than evaluating only on limited test data

2. Making the outputs of all systems publicly accessible so that anyone can run ensemble learning or other algorithms to create results better than the best single system

3. repeating the task with the larger and better training data from the output of the previous tasks

Project "SHINRA" started in 2017 with the aforementioned scheme, which we call "Resource by Collaborative Contribution (RbCC)" [SHINRA-HP]. The SHINRA tasks in 2018, 2019, and 2020 include the attribute extraction task in Japanese Wikipedia [Sekine et al., 2019]. Here, we report the first multi-lingual task and present future directions of the project.

The final goal of the SHINRA project is to create the structured knowledge base of Wikipedia, including the attribute. However, as a first step, we classify each Wikipedia entry into one or more ENE categories before extracting attribute values. The SHINRA2020-ML task resolves around classifying 30-language Wikipedia pages into ENE [ENE-HP] (ver.8.0) categories [SHINRA2020-ML-HP]. We classified most of the Japanese Wikipedia pages (920K pages) into the ENE categories already. We used language links to create the training data in 30 languages (for example, there are 275K language links from the Japanese

to German Wikipedia). So, the task is to categorize the remaining pages in 30 languages using the training data and evaluate the output based on a small portion of the pages that have no link from Japanese Wikipedia. This project is not only to compare the participating systems to see which system perform best, but also to create the knowledge base using outputs of the participated systems. We used the "ensemble learning technologies" to gather the outputs of the systems and create KB as accurately as possible.

## 2. Previous Work

Structured knowledge bases have been considered as one of the most important knowledge resources in the fields of Natural Language Processing. Previously, several major projects to construct structured knowledge bases have been conducted. One of the earliest projects is Cyc, and more recently are Wikipedia-based projects, such as DBpedia, YAGO, Freebase, and Wikidata. Moreover, some shared tasks aimed at building techniques for knowledge base structuring, such as KBP and CoNLL. Here, we introduce these resources and projects and describe the points we consider as issues to be solved in those projects.

Cyc ontology is a large knowledge base constructed as common sense knowledge[Lenat, 1995]. It was one of the large projects in AI between 1980 and 1990, which mainly used human labor to construct a knowledge base. The cost of construction and maintenance of handmade knowledge bases for the general domain was very high. The knowledge base suffered the "knowledge acquisition bottleneck" which includes problems in coverage and consistency.

DBpedia is a more recent project that constructs structured information from semi-structured data in Wikipedia, such as infoboxes and categories [Lehmann et al., 2015]. However, DBpedia is challenged with inaccuracy, low coverage, and lack of coherence. Like Cyc, infobox and categories in Wikipedia are also created by humans, however, they are non-experts of ontology. For example, we can easily notice that the categories are very noisy. In Japanese Wikipedia, "Shinjuku Station," which is a railway station, has a category "Odakyu Electric Railway", which is a major railway company using the station. However, a station is not an instance of a railway company, so this is not an appropriate category. There are so many examples like this in DBpedia. Also several inconsistencies are observed in the category structure, it is not even a hierarchy and there is a loop in the category structure. The attributes defined in DBpedia are not well organized in many categories.

YAGO is an ontology constructed by mapping Wikipedia articles to the WordNet synsets [Mahdisoltani et al., 2015]. Similar to DBpedia, YAGO adopts attribute information extracted from infoboxes because no attribute is defined in WordNet synsets.

Freebase is a project to construct a structured knowledge base using crowd-sourcing, similar to Wikipedia [Bollacker et al., 2008]. However, using the crowd-sourcing approach, Freebase does not have a well-organized ontology. It has many noises and lacks coherence since it was created using unorganized crowds. Currently, the Freebase project has been paused and integrated into Wikidata.

Wikidata aims to be a structured knowledge base, based on the crowd-sourcing scheme [Vrandečić and Krötzsch, 2014]. It has noises and lacks coherence since it was constructed by a bottom-up approach, similar to Wikipedia and Freebase. For example, by comparing the definition of a "city", "town" and "human settlements", we can easily observe inconsistencies

in the property (the numbers of properties are 30, 0 and 6, respectively), there are very biased properties such as "Danish urban area code" in "human settlements", there are many related ambiguous entities, such as "like a city", "city/town" and so on. Also, the category inconsistency is observed easily, for example, "city museum", "mayor" are subcategory of "city", although a mayor is not an instance of a "city." Wikipedia allows topics to be included in a category, however, this policy prevents making the category hierarchy a well-designed ontology.

KBP is a shared task organized by NIST for establishing a technology to construct a structured knowledge base from non-structured documents[U.S. National Institute of Standards and Technology (NIST) , 2018]. It consists mainly of two tasks, one is Entity Discovery and Linking (EDL) which involves finding and identifying an entity defined in DB from documents. The other is Slot Filling, which involves extracting attribute information of the entity. It is a competition-based project and not for resource creation purposes.

FIne Grained Entity Recognition (FIGER) is a project to identify 112 named entity classes that are finely defined, such as ENE, from documents[Ling and Weld, 2012]. The category of FIGER is biased, and it does not have attribute definitions for each category.

## 3. Data

This section introduces two datasets used for the SHINRA2020-ML task. One is the ENE definition which is an ontology for the named entity, and hence, Wikipedia entities are categorized. The other one is the categorized Japanese Wikipedia data, which will be used to create the training data for 30 languages using Wikipedia's language links.

### 3.1 Extended Named Entity

To construct a structured knowledge base useful for NLP applications, well-structured ontology is essential and must be designed in a top-down manner. The structures of the knowledge base in DBpedia, Freebase, and Wikidata were created by crowds in a bottom-up manner, and all have similar characteristics including inconsistent categories, imbalanced ontologies, and ad hoc attributes. This is majorly attributed to the bottom-up manner of their knowledge base design. A top-down strategy is essential to design the ontology and attributes consistently. For a top-down designed ontology for named entities, we employed the ENE [ENE-HP]. ENE is a named entity classification hierarchy that includes the attribute definition for each category ([Sekine et al., 2002], [Sekine and Nobata, 2004], [Sekine, 2008]). It includes 219 fine-grained categories of named entities in a hierarchy of up to four layers.

It contains not only the fine-grained categories of the typical NE categories, such as "city" and "lake" for "location", and "company" and "political party" for "organization", but also contains new named entity types such as "products", "event", and "natural object". These categories are designed to cover a wide range of entities in the general world, which are often mentioned in an encyclopedia and many other resources. Figure 1 shows the ENE definition, version 8.0. Attributes are designed so important attributes in the Wikipedia entities of each category are covered based on the investigation of sample entities. For example, the attributes for "airport" categories include the following: "Reading", "IATA code", "ICAO code", "nickname", "name origin", "number of users per year", "the year of

the statistic", "the number of airplane landings per year", "old name", "elevation", "big city nearby", "number of runaways". Please refer to the ENE homepage [ENE-HP] for the complete definitions.

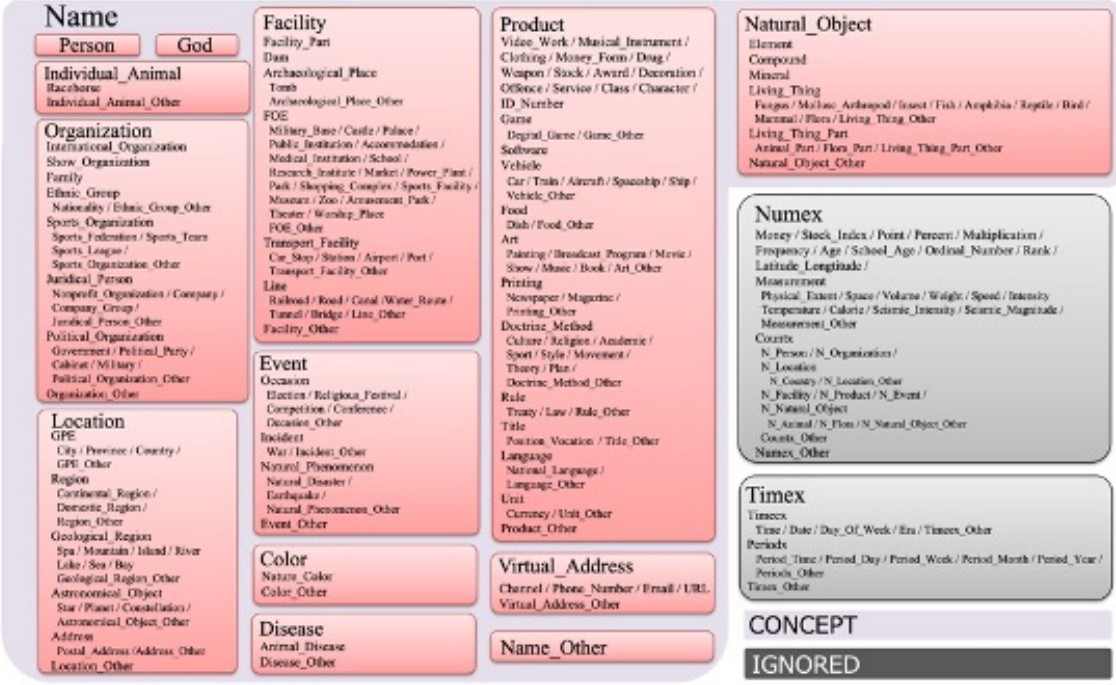

Figure 1: Definition of Extended Named Entity Hierarchy

### 3.2 Categorized Japanese Wikipedia

We categorized 920K Japanese Wikipedia pages into one or more of the 219 ENE categories before this project. For the categorization process, we excluded the less popular entities, i.e. having less than five incoming links (151K entities), and non-entity pages (about 53K pages) such as common nouns and simple numbers. This categorization was done using the machine learning method followed by hand-checking all data [Masatoshi et al., 2018]. The accuracy of the categorization was confirmed as 98.5%. The remaining 1.5% cover those that are ambiguous and are difficult even for human annotators, or for those having multiple categories. Table 1 shows the most frequent categories in the data of 920K pages.

### 4. Task Description

SHINRA2020-ML is the first shared-task of text categorization in the SHINRA project. It tackled the problem of classifying 30 language Wikipedia entities in fine-grained categories with 219 categories defined in ENE (ver. 8.0). Participants selected one or more target languages, and for each language, the system classified all Wikipedia pages in the target language(s). We provided the training data for 30 languages, created by the categorized

Table 1: Most Frequent Categories in Japanese Wikipedia

| Category | Freq. | Category | Freq. |
|---|---|---|---|
| Person | 247,983 | School | 23,609 |
| City | 45,306 | Literature | 18,515 |
| Artefact other | 33,453 | Movie | 17,901 |
| Broadcast Program | 32,050 | Train station | 15,863 |
| Company | 26,746 | Sports event | 15,863 |

Japanese Wikipedia of 920K pages and Wikipedia language links for 31 languages (including Japanese). Out of 2,263K German Wikipedia pages, 275K pages had a language links from Japanese Wikipedia, which served as silver (i.e. bit noisy) training data for German. Thus, the task became "to classify the remaining 1,988K pages into 219 categories, based on the training data" (actually, the participants are requested to categorize the training data with their system, as well). The same holds for the other 29 languages, as shown in the data statistics shown in Table 2.

### 4.1 Schedule

SHINRA2020-ML has been conducted according to the schedule listed in Table:3.

### 4.2 Participants

Ten groups from Seven countries participated in this study. The list of participant groups and the task they participated in are shown in Table 4. The description of most of systems are reported in the NTCIR-15 conference [Bui and Le-Hong, 2020] [Cardoso et al., 2020] [Abhishek et al., 2020] [Nakayama and Sekine, 2020] [Yoshioka and Koitabashi, 2020] [Nishikawa and Yamada, 2020] [Yoshikawa et al., 2020].

### 4.3 Evaluation Results

For each target language, a group can submit up to three runs based on different methods. A system classifies each page into one or more of the ENE (ver.8.0) categories correctly. The evaluation pages are selected from those not linked from Japanese pages. The selection method for the evaluation data is not disclosed since we can compare future results using the same data. Suppose the estimated category is not an exact match, the system get no score for that output. We evaluated the performance of systems on a multi-label classification using the micro average F1 measure, i.e. the harmonic mean of precision and micro-averaged recall. Note, the distribution of the category in the test data may differ from that of target or training data.

Table 5 shows the F-measure of participating systems. "Late submission" indicates those who submit the results after the deadline.

### 4.4 Results on RbCC

We tried a simple ensemble learning, called the majority voting. We used systems from all but one participants, which claims that they are not aiming at building a good performance

Table 2: Wikipedia Statistics in 31 languages

| Language | num. of pages | Links from ja | Ratio |
|---|---|---|---|
| English (en) | 5,790,377 | 439,354 | 7.6 |
| Spanish (es) | 1,500,013 | 257,835 | 17.2 |
| French (fr) | 2,074,648 | 318,828 | 15.4 |
| German (de) | 2,262,582 | 274,732 | 12.1 |
| Chinese (zh) | 1,041,039 | 267,107 | 25.7 |
| Russian (ru) | 1,523,013 | 253,012 | 16.6 |
| Portuguese (pt) | 1,014,832 | 217,896 | 21.5 |
| Italian (it) | 1,496,975 | 270,295 | 18.1 |
| Arabic (ar) | 661,205 | 73,054 | 11.0 |
| Japanese | 1,136,222 | – | – |
| Indonesian (id) | 451,336 | 115,643 | 25.6 |
| Turkish (tr) | 321,937 | 111,592 | 34.7 |
| Dutch (nl) | 1,955,483 | 199,983 | 10.2 |
| Polish (pl) | 1,316,130 | 225,552 | 17.1 |
| Persian (fa) | 660,487 | 169,053 | 25.6 |
| Swedish (sv) | 3,759,167 | 180,948 | 4.8 |
| Vietnamese (vi) | 1,200,157 | 116,280 | 9.7 |
| Korean (ko) | 439,577 | 190,807 | 43.7 |
| Hebrew (he) | 236,984 | 103,137 | 43.5 |
| Romanian (ro) | 391,231 | 92,002 | 23.5 |
| Norwegian (no) | 501,475 | 135,935 | 27.1 |
| Czech (cs) | 420,195 | 135,935 | 25.1 |
| Ukrainian (uk) | 881,572 | 181,122 | 20.5 |
| Hindi (hi) | 129,141 | 30,547 | 23.6 |
| Finnish (fi) | 450,537 | 144,750 | 32.1 |
| Hungarian (hu) | 443,060 | 120,295 | 27.2 |
| Danish (da) | 242,523 | 91,811 | 35.6 |
| Thai (th) | 129,294 | 59,791 | 46.2 |
| Catalan (ca) | 601,473 | 139,032 | 23.1 |
| Greek (el) | 157,566 | 60,513 | 38.4 |
| Bulgarian (bg) | 248,913 | 89,017 | 35.7 |

system, but aiming to show the degree of usefulness of Wikipedia category for this task. The results are shown in Figure 6. In this table, languages that achieved better results by majority voting than the single best system are marked green. Seventeen languages which got the better results, indicating a great performance for the RbCC scheme. In the table, "oracle F1" denotes the F1 score for selecting the correct answer from the pool of outputs by all systems.

Table 3: SHINRA2020-ML Schedule

| Date | Event |
|---|---|
| January, 2020 | Data release |
| April, 2020 | Homepage and CFP open |
| August 31, 2020 | Registration and Result submission deadline |
| September 16, 2020 | Evaluation results due back to participants |
| Dec 8-11, 2020 | NTCIR-15 Conference |

Table 4: SHINRA2020-ML Participants

| Group ID | Country | Participated Language |
|---|---|---|
| CMVS | Finland | 1 (ar) |
| FPTAI | Vietnam | 30 (all) |
| HUKB | Japan | 30 (all) |
| PribL | Portugal | 15 (ar, cs, de, en, es, fr, it, ko, nl, no, pl, pt, ru, tr, zh) |
| RH312 | India | 6 (bg, fr, hi, id, th, tr) |
| TKUIM | Taiwan | 30 (all) |
| Ousia | Japan | 9 (ar,de,es,fr,hi,it,pt,th,zh) |
| Uomfj | Australia/Japan | 28 (except for el, sv) |
| Vlp | Vietnam | 1 (vi) |
| LIAT | (organizer) Japan | 30 (all) |

## 5. SHINRA tasks

Project SHINRA has been ongoing since 2017. The tasks are not limited to only categorizing multi-lingual Wikipedia, but also entail the attribute extraction and linking tasks for Japanese Wikipedia, as shown in Table 7. We are conducting two tasks in 2021 [SHINRA-HP].

## 6. Conclusion

This paper report progress from the ongoing SHINRA project, particularly, the SHINRA2020-ML task. We proposed a knowledge base creation scheme, "Resource by Collaborative Contribution (RbCC)". The SHINRA2020-ML task categorizes 30-language Wikipedias into Extended Named Entity (ENE), the top-down definition of NE categories. Ten groups participated in the task, and the ensemble learning results show that the RbCC scheme is promising. Furthermore, we plan to conduct the SHINRA2021-ML and SHINRA2021-LinkJP tasks based on the RbCC scheme. We express our deep appreciation to all participants and collaborators who helped this project. With many participation, we can try ensemble learning and achieve the goal. We hope to expand the idea of the RbCC scheme to other ML tasks in general, not limited to sHINRA or similar task and resource.

## Acknowledgments

Table 5: SHINRA2020-ML task Result

| | Group ID | FPTAI | LIAT | PribL | PribL | RH312 | ousia | uomfj | uomfj | uomfj | FPTAI | HUKB | HUKB | HUKB | LIAT |
|---|---|---|---|---|---|---|---|---|---|---|---|---|---|---|---|
| | Method ID | BERT | ML-BERT | BERTGRU | BERTLIN CONCAT | RnnGnnXlmr | RoBERTa+wiki2vec+wikidata | jointrep | jointrepPostprocess | jointrepUnionPostprocess | BERT | AB | ABC | AC | ML-BERT |
| | Late Submission | | | | | | | | | | Y | Y | Y | Y | Y |
| ar | Arabic | 73.25 | 63.16 | 76.27 | 75.45 | - | 70.52 | 64.55 | 64.55 | 64.55 | 73.25 | 30.98 | 30.98 | 13.51 | - |
| bg | Bulgarian | 83.77 | 75.20 | - | - | 82.13 | - | 83.07 | 83.07 | 83.07 | 83.28 | 60.86 | 61.06 | 28.09 | - |
| ca | Catalan, Valencian | 52.55 | 76.28 | - | - | - | - | 79.82 | 79.82 | 79.82 | 81.10 | 42.34 | 42.54 | 16.26 | - |
| cs | Czech | 84.47 | 79.46 | - | 81.19 | - | - | 81.29 | 81.29 | 81.29 | 83.74 | 52.61 | 52.61 | 18.86 | - |
| da | Danish | 82.30 | 74.80 | - | - | - | - | 80.56 | 80.56 | 80.56 | 81.74 | 49.01 | 49.01 | 13.99 | - |
| de | German | 22.62 | 79.49 | 80.24 | 79.83 | - | 81.86 | 81.03 | 81.03 | 81.03 | 81.26 | 53.72 | 53.82 | 26.81 | - |
| el | Greek, Modern (1453-) | 84.40 | 72.43 | - | - | - | - | - | - | - | 84.10 | 7.51 | 7.51 | 7.51 | - |
| en | English | 82.23 | 78.56 | 81.27 | 80.12 | - | - | 82.73 | 82.57 | 82.68 | 81.96 | 45.11 | 45.11 | 11.92 | - |
| es | Spanish, Castilian | 80.60 | 77.73 | 80.30 | 80.72 | - | 80.94 | 81.39 | 81.39 | 81.39 | 80.60 | 49.21 | 49.11 | 19.50 | - |
| fa | Persian | 81.70 | 75.42 | - | - | - | - | 80.38 | 80.38 | 80.38 | 81.52 | 45.59 | 45.59 | 15.66 | - |
| fi | Finnish | 83.62 | 79.13 | - | - | - | - | 80.91 | 80.91 | 80.91 | 83.36 | 53.15 | 53.45 | 17.06 | - |
| fr | French | 21.59 | 76.88 | 77.93 | 78.52 | 80.31 | 81.01 | 78.21 | 78.21 | 78.21 | 80.68 | 43.84 | 43.74 | 11.23 | - |
| he | Hebrew | 83.79 | 79.11 | - | - | - | - | 81.09 | 81.09 | 81.09 | 84.21 | 59.95 | 60.05 | 15.78 | - |
| hi | Hindi | 76.43 | 16.49 | - | - | 71.70 | 69.75 | 66.67 | 66.67 | 66.67 | 75.65 | 39.70 | 39.51 | 22.02 | - |
| hu | Hungarian | 85.46 | 78.93 | - | - | - | - | 85.02 | 85.02 | 85.02 | 84.78 | 69.15 | 69.44 | 26.09 | - |
| id | Indonesian | 81.93 | 72.45 | - | - | 77.56 | - | 78.51 | 78.51 | 78.51 | 81.65 | 44.07 | 44.47 | 16.28 | - |
| it | Italian | 26.55 | 81.36 | 81.92 | 81.89 | - | 81.21 | 82.02 | 82.02 | 82.02 | 82.81 | 45.55 | 45.55 | 12.06 | - |
| ko | Korean | 83.67 | 80.38 | 81.51 | 81.04 | - | - | 82.51 | 82.51 | 82.51 | 83.77 | 63.68 | 63.98 | 13.95 | - |
| nl | Dutch, Flemish | 83.29 | 79.86 | 80.95 | 81.26 | - | - | 81.64 | 81.64 | 81.64 | 83.17 | 42.36 | 42.45 | 17.12 | - |
| no | Norwegian | 80.53 | 76.50 | - | 78.39 | - | - | 78.79 | 78.79 | 78.79 | 80.17 | 34.58 | 34.58 | 11.33 | - |
| pl | Polish | 84.53 | 80.60 | 82.73 | 83.46 | - | - | 84.52 | 84.52 | 84.52 | 84.07 | 62.72 | 63.51 | 32.55 | - |
| pt | Portuguese | 83.23 | 78.49 | 82.36 | 81.88 | - | 81.40 | 80.87 | 80.87 | 80.87 | 82.70 | 42.32 | 42.62 | 16.10 | - |
| ro | Romanian, Moldavian, Moldovan | 84.60 | 76.17 | - | - | - | - | 80.83 | 80.83 | 80.83 | 84.60 | 57.60 | 57.70 | 28.50 | - |
| ru | Russian | 84.08 | 79.09 | 82.60 | 83.07 | - | - | 82.90 | 82.90 | 82.90 | 83.44 | 42.04 | 42.24 | 11.30 | - |
| sv | Swedish | 83.18 | 71.63 | - | - | - | - | - | - | - | 83.44 | 50.32 | 50.62 | 21.98 | 79.58 |
| th | Thai | 81.26 | 49.58 | - | - | 76.77 | 76.36 | 65.02 | 65.02 | 65.02 | 81.16 | 39.98 | 40.38 | 24.05 | - |
| tr | Turkish | 86.50 | 77.19 | 84.36 | 83.23 | 83.28 | - | 84.85 | 84.85 | 84.85 | 86.03 | 61.88 | 62.48 | 16.73 | - |
| uk | Ukrainian | 83.12 | 78.71 | - | - | - | - | 81.61 | 81.61 | 81.61 | 82.61 | 60.29 | 60.19 | 22.51 | - |
| vi | Vietnamese | 80.34 | 75.24 | - | - | - | - | 77.06 | 77.06 | 77.06 | 80.42 | 60.38 | 60.48 | 22.14 | - |
| zh | Chinese | 81.25 | 77.97 | 78.38 | 79.37 | - | 79.76 | 78.58 | 78.58 | 78.58 | 80.60 | 21.22 | 21.42 | 17.57 | - |

Table 6: Ensemble Learning Result

| ISO 639-1 | Language | Group ID | Method | Precision | Recall | F1 | Majority Voting F1 | Oracle F1 | Num Groups | Num Methods |
|---|---|---|---|---|---|---|---|---|---|---|
| tr | Turkish | FPTAI | BERT | 84.22 | 88.92 | 86.50 | 87.38 | 92.71 | 7 | 12 |
| hu | Hungarian | FPTAI | BERT | 82.89 | 88.19 | 85.46 | 85.49 | 91.18 | 5 | 9 |
| ro | Romanian, Moldavian, Moldovan | FPTAI | BERT | 81.40 | 88.07 | 84.60 | 84.47 | 91.97 | 5 | 9 |
| pl | Polish | FPTAI | BERT | 82.01 | 87.22 | 84.53 | 85.27 | 91.55 | 6 | 11 |
| cs | Czech | FPTAI | BERT | 81.31 | 87.88 | 84.47 | 84.52 | 90.59 | 6 | 10 |
| el | Greek, Modern (1453-) | FPTAI | BERT | 81.34 | 87.70 | 84.40 | 75.76 | 90.26 | 4 | 6 |
| he | Hebrew | FPTAI | BERT | 80.50 | 88.28 | 84.21 | 84.34 | 92.22 | 5 | 9 |
| ru | Russian | FPTAI | BERT | 81.59 | 86.73 | 84.08 | 84.73 | 90.50 | 6 | 11 |
| bg | Bulgarian | FPTAI | BERT | 80.94 | 86.81 | 83.77 | 84.74 | 91.04 | 6 | 10 |
| ko | Korean | FPTAI | BERT | 80.44 | 87.39 | 83.77 | 84.22 | 91.95 | 6 | 11 |
| fi | Finnish | FPTAI | BERT | 79.98 | 87.61 | 83.62 | 83.61 | 90.46 | 5 | 9 |
| sv | Swedish | FPTAI | BERT | 80.20 | 86.94 | 83.44 | 82.21 | 91.38 | 5 | 9 |
| nl | Dutch, Flemish | FPTAI | BERT | 81.27 | 85.41 | 83.29 | 83.85 | 90.73 | 6 | 11 |
| pt | Portuguese | FPTAI | BERT | 79.80 | 86.97 | 83.23 | 83.98 | 93.17 | 7 | 12 |
| uk | Ukrainian | FPTAI | BERT | 80.05 | 86.43 | 83.12 | 83.92 | 89.81 | 5 | 9 |
| it | Italian | FPTAI | BERT | 79.98 | 85.84 | 82.81 | 83.72 | 92.77 | 7 | 12 |
| en | English | uomfj | jointrep | 81.77 | 83.71 | 82.73 | 82.66 | 89.60 | 6 | 11 |
| da | Danish | FPTAI | BERT | 79.47 | 85.33 | 82.30 | 80.93 | 90.49 | 5 | 9 |
| id | Indonesian | FPTAI | BERT | 78.23 | 86.01 | 81.93 | 81.44 | 90.40 | 6 | 10 |
| de | German | ousia | RoBERTa+wiki2vec+wikidata | 82.59 | 81.15 | 81.86 | 82.45 | 90.63 | 7 | 12 |
| fa | Persian | FPTAI | BERT | 79.35 | 84.18 | 81.70 | 81.09 | 88.54 | 5 | 9 |
| es | Spanish, Castilian | uomfj | jointrepUnionPostprocess | 82.20 | 80.59 | 81.39 | 82.88 | 89.25 | 7 | 12 |
| th | Thai | FPTAI | BERT | 78.07 | 84.72 | 81.26 | 81.14 | 90.69 | 7 | 11 |
| zh | Chinese | FPTAI | BERT | 78.83 | 83.82 | 81.25 | 80.83 | 89.45 | 6 | 11 |
| ca | Catalan, Valencian | FPTAI | BERT | 77.34 | 85.25 | 81.10 | 80.57 | 91.11 | 5 | 9 |
| fr | French | ousia | RoBERTa+wiki2vec+wikidata | 81.09 | 80.93 | 81.01 | 81.92 | 90.32 | 8 | 13 |
| no | Norwegian | FPTAI | BERT | 77.58 | 83.71 | 80.53 | 81.27 | 89.44 | 6 | 10 |
| vi | Vietnamese | FPTAI | BERT | 77.61 | 83.43 | 80.42 | 80.16 | 91.62 | 6 | 10 |
| hi | Hindi | FPTAI | BERT | 73.67 | 79.41 | 76.43 | 73.67 | 84.51 | 7 | 11 |
| ar | Arabic | PribL | BERTGRU | 76.80 | 75.74 | 76.27 | 73.39 | 90.89 | 8 | 13 |
| | | MAX | | 84.22 | 88.92 | 86.50 | 87.38 | 93.17 | 8 | 13 |
| | | MIN | | 73.67 | 75.74 | 76.27 | 73.39 | 84.51 | 4 | 6 |

This work was supported by JSPS KAKENHI Grant Number JP20269633.

Table 7: SHINRA tasks

| Task | Description |
| --- | --- |
| SHINRA2018 | Extracting attribute values in 5 categories in Japanese |
| SHINRA2019 | Extracting attribute values in 30 categories in Japanese |
| SHINRA2020-JP | Extracting attribute values in 45 categories in Japanese |
| SHINRA2020-ML | Categorizing 30-language Wikipedias |
| SHINRA2021-LinkJP | Link Attribute values to corresponding Wikipedia pages |
| SHiNRA2021-ML | Categorizing 30-language Wikipedias |

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
