# OpenReview forum: "SHINRA2020-ML: Categorizing 30-language Wikipedia into fine-grained NE based on "``Resource by Collaborative Contribution" scheme"
_AKBC.ws/2021/Conference — AKBC 2021_

### Official Review · Reviewer_9RQv · 2021-07-22
**Well-motivated problem, limited technical depth, important details are missing.**

**Rating:** 6
**Confidence:** 3

**Review:**

This paper describes a system that aims to categorize 30-language Wikipedia
pages into a structured knowledge base. As opposed to previous work, the knowledge base is defined in a top-down manner. Importantly, the output of the proposed system is publicly available and thus may be used by other systems/applications in the future for multiple NLP tasks.

Strengths:
1. The contribution of this system is well defined w.r.t. other publicly available knowledge bases, and the authors provide an extensive explanation regarding the existing literature.

2. This paper provides a publicly available well-structured knowledge base. Other systems and applications may use that in the future in multiple NLP tasks.

3. The authors have proposed a scheme of knowledge base creation named ``Resource by Collaborative Contribution (RbCC)", showing it to work well in practice

Weaknesses:

1. Even though the task at hand is well motivated and the contribution of this project is significant, the technical depth of this paper is limited. The paper contains only a report regarding an ongoing project.

2. Important details are missing. Who are the participant groups? How did you recruit them? Which method did each group use? What is their background? Another important piece of information that is missing is regarding the ground truth data (i.e., the categorization of the Japanese Wikipedia pages). Which ML methods were used? Are groups using similar or the same methods succeeded better than others? It would be interesting to discuss this issue in more detail.

3. A natural question here is whether they should have used a majority vote or not. Perhaps there is a significant difference in the performance of each group, and a weighting strategy would be more appropriate. This choice of using majority vote as the aggregation function should be supported by an experiment, showing it outperforms alternative baselines.

4. It would be interesting to demonstrate the contribution of this project by, e.g., evaluating the performances of different ML tasks executed over the constructed KB and other KBs. This will support the claim that a KB built in a top-down manner is better than o one built in a bottom-up fashion, as the authors have claimed.

---

### Official Review · Reviewer_toGH · 2021-07-22
**Resource Paper, Lack of novelty and technicality**

**Rating:** 5
**Confidence:** 3

**Review:**

The paper reports a shared task of categorizing the Wikipedia pages into fined-grained named entity categories in 30-languages.
It follows the RbCC scheme and AKBC system to construct a KB. They provided an Extended Named Entity hierarchy for labeling Wikipedia pages. \
The research question is both relevant and interesting, and the findings seem reasonable. However, the explanation and presentation of the method, experiments, and results are lacking.

Strength:
1. Publicly available resources that can be beneficial for multi-lingual NLP tasks.
 2. Empirical results showing the RbCC scheme is useful in KB construction.

Weaknesses:
1. The Extended Named Entity hierarchy which is base knowledge to categorized pages is not well defined. The definition of Named Entity is wage. How does the ENE align with other ontologies such as UFO, BFO?
2. The details about the shared task, such as participants' methods and error analysis, are missing?

---

### Official Review · Reviewer_PCvS · 2021-07-22
**Report of collaborative effort to classify Wikipedia pages in 30 new languages.**

**Rating:** 7
**Confidence:** 4

**Review:**

This paper presents the design and results of a shared task that fosed on categorizing Wikipedia entity pages in 30 languages. The task attracted 10 participants, working on overlapping subsets of the target languages, and the output predictions will be shared with the community as a knowledge resource.

The task requires systems to categorize pages into the curated ENE type set. This type set is motivated as being better defined than the user-generated categories present in Wikipedia and a high quality classifier has already been created for Japanese. This Japanese classifier is used to label 960k pages and provide distant labeling for associated pages in the 30 target languages. Submissions operate directly on the target languages and should be able to extend the categorization to pages that are not present in the Japanese Wikipedia.

Results are presented for an ensemble of the best systems. This ensemble usually outperforms the single best submission by a small amount and F1 scores exceed 80% for almost all languages. I didn't understand the 'oracle F1' score, which is significantly higher, and this should be explained in the paper (does the oracle choose a best system for a page, or something else?)

Strengths:
- Nice collaborative effort that engaged a diverse set of teams.
- Cretes type classification of Wikipedia that will be released and could be useful for downstream tasks.

Weaknesses:
- Evaluated against predictions that are already available for Japanese Wikipedia. Evaluation would be stronger if it investigated pages present in the target language but not present in the (already classified) Japanese Wikipedia.

---

### Decision · Program_Chairs · 2021-08-18

**Decision:**

Accept

**Comment:**

This paper presented a shared task that focused on categorizing the Wikipedia pages into fine-grained named entity categories in 30-languages. We all agreed that this work contributed a publicly available knowledge base resource, and its research question is relevant to the community. Reviewers also pointed out a few weaknesses, including annotation details, the paper's contribution/technical depth, the implication/impact of the constructed KB on downstream applications. With the authors’ responses, I think the pros seem to outweigh the cons of this work a bit, and these concerns can be further explained/clarified in a revised version. I urge authors to incorporate reviewers’ suggestions and their planned revisions into the final version of this work.